# Miniaturised Infrared Spectrophotometer for Low Power Consumption Multi-Gas Sensing

**DOI:** 10.3390/s20143843

**Published:** 2020-07-09

**Authors:** Manu Muhiyudin, David Hutson, Desmond Gibson, Ewan Waddell, Shigeng Song, Sam Ahmadzadeh

**Affiliations:** Scottish Universities Physics Alliance, Institute of Thin Films, Sensors and Imaging, University of the West of Scotland, Paisley PA1 2BE, UK; Manu.Muhiyudin@meggitt.com (M.M.); des.gibson@uws.ac.uk (D.G.); ewan.waddell@btinternet.com (E.W.); shigeng.song@uws.ac.uk (S.S.); sam.ahmadzadeh@uws.ac.uk (S.A.)

**Keywords:** lead selenide, micro hotplate, linear variable filter, fixed line pass band filters, photo detection, non-dispersive infrared spectroscopy

## Abstract

Concept, design and practical implementation of a miniaturized spectrophotometer, utilized as a mid-infrared-based multi gas sensor is described. The sensor covers an infrared absorption wavelength range of 2.9 to 4.8 um, providing detection capabilities for carbon dioxide, carbon monoxide, nitrous oxide, sulphur dioxide, ammonia and methane. A lead selenide photo-detector array and customized MEMS-based micro-hotplate are used as the detector and broadband infrared source, respectively. The spectrophotometer optics are based on an injection moulded Schwarzschild configuration incorporating optical pass band filters for the spectral discrimination. This work explores the effects of using both fixed-line pass band and linear variable optical filters. We report the effectiveness of this low-power-consumption miniaturized spectrophotometer as a stand-alone single and multi-gas sensor, usage of a distinct reference channel during gas measurements, development of ideal optical filters and spectral control of the source and detector. Results also demonstrate the use of short-time pulsed inputs as an effective and efficient way of operating the sensor in a low-power-consumption mode. We describe performance of the spectrometer as a multi-gas sensor, optimizing individual component performances, power consumption, temperature sensitivity and gas properties using modelling and customized experimental procedures.

## 1. Introduction

Non-dispersive infrared (NDIR) sensors are based on mid-infrared absorption spectroscopy [1,2], providing benefits such as cost and sensitivity [1,2,3,4], integration as a miniaturised sensor configuration compared to other optical-based gas-sensing techniques [5] such as tunable diode laser absorption spectroscopy [6], photoacoustic spectroscopy [7] and quartz-enhanced photoacoustic spectroscopy [8].

Moreover, NDIR sensors are emerging as a replacement for chemical-based gas sensors [9,10,11,12,13], addressing inherent problems with chemical sensors such as short lifespan due to gradual degradation of the catalyst, high power consumption, susceptibility to gas poisoning and consequent non-fail to safe [9]. The NDIR-based gas sensor described in this work addresses these shortfalls.

Non-Dispersive techniques are finding extensive applications in different fields such as environmental gas monitoring [14], industrial safety [14], medical diagnostics [14,15,16,17], demand control ventilation in building air handling systems [18,19,20], agri-tech [21,22]. NDIR sensors are based on different gases absorbing infrared radiation at uniquely defined wavelengths [22]. When compared with available electrolytic gas detectors, NDIR sensors can provide enhanced accuracy, operate at lower power, are not prone to poisoning and have “fail to safe” performance. They also have excellent response times (time constants < 1 s possible) and excellent sensitivities to gas. The zero drift issues [23] associated with semiconductor gas sensors and problems such as cross-sensitivity, drift and short life-spans associated with electrochemical gas sensors, make optical absorption-based gas sensors an attractive prospect for gas-sensing applications.

Realisation of an NDIR sensor based on spectral differentiation requires the following—(1) a mid-infrared source providing broadband radiation covering the wavelength range of interest, achieved by the use of a blackbody source emitting a broadband infrared spectrum; (2) an optical filter discriminating specific wavelengths from the incident radiation (the present work explores the usage of both fixed line pass band optical filters and linear variable filters for achieving spectral selection); (3) the ability to modulate the source to maximise signal/noise ratio; (4) a photo detector sensitive to IR radiation in the spectral range covering absorption wavelengths of the gases of interest.

The single and multi-gas sensor described here has been developed for mass production, using injection-moulded reflective optics and CMOS-based light source and detector manufacture, combined with miniaturized electronics, reducing overall size of the actual multi-gas sensor size to within 25 × 25 × 25 mm.

## 2. Materials and Methods

### 2.1. Optical Design and Construction

An optical design based on a Schwarzschild configuration [24] magnifies the optical source output, matching the spatial input to a lead selenide array of detectors [25]. An optical filter (described in detail in Section 2.4) placed on the detector array provides spectral discrimination across the array elements. A basic representation of the sensor is provided in Figure 1.

An optical light source is a spatially extended micro-hotplate with a 0.6 × 0.2 mm active area size (Section 2.2), magnifies source output by 10 and 5, respectively, in length and width, respectively, to 6 × 1.0 mm, matching the detector array size. Overall gas absorption pathlength is 60 mm (achieves required absorption over a wide range of gases—see Section 2.6), comprising a three-piece injection moulded (blend of polycarbonate and platable acrylonitrile butadiene styrene) folded optic, as shown in Figure 1b. All reflective surfaces are sputter gold coated, providing high mid infrared reflectivity (typically 98% per surface). Figure 1c shows the actual three pieces injection moulded assembly beside a United Kingdom 50 pence piece (diameter 23 mm). Overall, sensor assembly dimensions are 25 × 25 × 25 mm, providing a miniaturised NDIR multi-gas sensor compared with other optical gas-sensing approaches [3,4,5].

### 2.2. MEMS Micro-Hotplate as the Infrared Source

The micro-hotplate [26,27,28] used here is constructed using polysilicon (Poly-Si) thin-film sandwiched between two silicon nitride substrates. Aluminum metal is used as the conducting electrode to the micro-heater. Figure 2a shows the electrodes connecting to the polysilicon membrane showing the cut-out sections or slits. When electrically energized, the micro-heater initiates warm-up due to electro-thermal interaction. There will be significant conduction losses from the body to the substrate and also some convection losses [29]. Energy will also be radiated from the surface, providing an effective blackbody (broadband) infrared source. Figure 2b shows the intensity distribution of the emitted radiation.

The ideal micro-hotplate characteristics are low thermal mass, temperature uniformity, low power consumption and resistance to low pressure and high temperatures.

Non-uniform voltage equipotential lines near edges of the slits cause a greater pull of electrons and consequently, higher current density. This results in increased temperature between the slits. The micro-hotplate length is customized to 0.6 mm length, providing magnification to 6 mm at the detector array (see Section 2.3). The broadband spectrum produced by the micro hotplate provides the required signal spread over all elements of the lead selenide infrared detector array. The hotplate is operated in a constant voltage mode, using a circuit developed specifically to provide a battery compatible electromotive force of <5 V.

A future publication will describe the MEMS hotplate design and fabrication in detail.

### 2.3. Infrared Detector

Photoconductive lead selenide is used as the infrared detector, in the form of an 8 elementarray −1.0 mm × 0.8 mm for each pixel of the array. The temperature sensitivity of lead salt detectors [30,31] requires real-time temperature compensation, achieved using a built-in reference channel.

With the optical filter(s) integrated onto the detector array, the drop in output of specific channel(s) can be made to correspond to specific gas detection, as shown in Figure 3. The array element with zero change in output due to infrared absorption is assigned for the reference channel. The reference channel is used for drift compensation due to temperature changes and path length variation. This improves sensor accuracy and repeatability.

A 60 mm path length demonstrator developed with transmissive bulk optics is shown in Figure 4a and Figure 5. A single-element lead selenide detector was used during testing, with a micro hotplate point source providing the input infrared radiation at 4 Hz and 12 Hz frequencies. The response of the system was recorded at a time constant of 12 ms.

### 2.4. Optical Filter

Both commercial as well as in-house manufactured fixed filters were used as spectral differentiators for the experimental needs. Modelling (Section 2.6) describes the ideal pass band characteristics of the filter for the optimal performance during gas detection.

#### 2.4.1. Linear Variable Filters

Linear variable filters (LVF) are spatially variable optical band pass filters, in which the center wavelength changes with position in a linear manner along one direction of the filter. The LVF used in this work was fabricated at the University of the West of Scotland using a unique and volume-scalable technology [32,33,34]. The manufactured LVF has a spectral range of 2.9 to 4.8 µm with a pass band width of 1.5%. and peak transmission of 70%, thereby providing wavelength discrimination across the detector array and covering the range of gases of interest (e.g., CH_4_, N_2_O, CO_2_, CO).

#### 2.4.2. Fixed-Line Optical Pass Band Filter

Optical pass band filters transmit only a particular band of wavelength and blocks the other wavelengths on either side of the band. The band pass can be anywhere from less than an Ångström to a few hundred nanometers and such filters are usually manufactured by combining a long pass and a short pass optical filter. Both commercial and in-house manufactured optical pass band filters were used for this work.

### 2.5. Electronics

As described in Section 2.1 the injection-molded optics use a Schwarzschild-based optical dome design [24] where the source is magnified, matching the detector array spatial configuration. Associated electronics controls the micro hotplate input and up to 8 detector channels. A 16-bit ADC and 12-bit DAC are used for measurements and temperature offset corrections, respectively. An ultra-low power FPGA is utilized in the design, which is used to control the A-to-D converter, D-to-A converter, input multiplexer and also the hot plate pulsing. The processor clocks the FPGA at 42 MHz, which manages the system timing. A GUI is developed which can set the pulse-width, pulse-decay time, collect ambient temperature measurements and the offset measurements (which is a direct measure of the resistance change in the detector channel due to temperature sensitivity). The hot plate is pulsed by a constant voltage driver circuit and mirrors the processor DAC output voltage. The data is read out through a serial interface.

### 2.6. Modelling

Mathcad 15.0 is used for the modelling and provides a convenient and dynamic platform to approach the complex modelling of the sensor. The sensor possesses a net optical path length of 60 mm. Gas absorption data are imported into the MATHCAD model from the HITRAN 2012 data base in Spectral Calc online spectral modelling tool. A total of 600 data points of wavelength and absorption strength data for each of the gases are extracted locally for modelling.

A MEMS-based micro hotplate with doped polysilicon as the active heating element is modeled as the broadband infrared source. The dimension of the polysilicon layer is 600 × 650 × 1 µm, which is sandwiched between layers of Silicon Nitride and Silicon Dioxide substrates. By Planck’s law for electromagnetic radiation, the net spectral radiant exitance or the net black body radiation emitted into a hemisphere by the micro hotplate at a certain temperature T and wavelength λ, in power per unit area per wavelength interval is given by:(1)$$W_{\mathrm{Planck}}\left(\lambda ,T\right)=\frac{2\pi hc^{2}}{\lambda^{5}}{\left(e^{\left(\left(\frac{hc}{\lambda kT}\right)-1\right)}\right)}^{-1}$$
where h is the Planck’s constant, c is the speed of light, k is the Boltzmann’s constant and λ the wavelength of the electromagnetic radiation. The net source spectral radiant power is then estimated at a specific source temperature T_S_, by integrating the total black body radiation over the whole area of the micro hot plate. It is modeled as shown below in Figure 6.

The photoconductive Lead Selenide detector is modeled for its specific detectivity (D*) within the MWIR detection range of 2 to 5 um. The Lead Selenide detector is used in the form of an 8-detector array—1.0 mm × 0.8 mm for each element of the array. Lead Selenide is highly temperature-sensitive and one of the detector array elements is designated as the reference channel to compensate for temperature drifts and cross-sensitivities during gas measurements. Noise factors such as Johnson–Nyquist noise and dark current noise are also incorporated into the detectivity estimation.

Specific detectivity (D*) for a wavelength λ and detector temperature T is estimated as:(2)$${D^{*}}_{\lambda T}\left(\lambda ,T\right)=D_{\mathrm{PR}}\times S_{\mathrm{PD}}\left(\lambda ,T\right)$$

DPR is a parameter dependent on the noise spectral density of the sensor, area of the detector array element and the detection bandwidth. $S_{PD}\left(\lambda ,T\right)$ is a function of the responsivity of the detector. The detector responsivity, is modelled as:(3)$$S_{0PD^{\left(T\right)}}=\frac{\eta_{\mathrm{detPD}}\times q_{e}\times G_{\mathrm{pc}}\left(T\right)}{h\times c}$$
where η is the quantum detector efficiency, which is the ratio of the number of electrons collected by the photoconductor to the total number of incident photons, q_e_ is the charge of an electron, G_pc_(T) is the photoconductive gain at temperature T, h is the Planck’s constant and c is the speed of light. G_pc_ is not necessarily 1. This is strongly temperature-dependent, and the power temperature dependence shown below is on the basis of temperature dependence of mobility, due to optical phonons which are usually dominant at moderate temperatures.

The Detectivity D* at 25 °C for a specific detector temperature T_D_ is thus given by:(4)$$D^{*}25\left(\lambda ,T_{D}\right)=\frac{{D^{*}}_{\lambda T}\left(\lambda ,T_{D}\right)}{N_{D^{*}25}}$$
where $N_{D^{*}25}$ is the Noise Spectral Density at 25 °C.

The net detectivity will be the area under the curve, integrating it over the detectable wavelength range. At 25 °C, the detectivity D* is calculated as 2.939 × 10^9^ Jones (slightly higher than the datasheet value of 2 × 10^9^ Jones for the commercial Lead Selenide detector array used for testing).

The detector response is modeled for its detectivity D*, as shown in Figure 7 below.

The variation of detectivity D* with temperature is then modelled for 0 °C, 25 °C and 50 °C, as shown in Figure 7. The ability to achieve spectral control for both the source and the detector is critical for ensuring accurate compensation while detecting gas. The source-detector product is of particular importance and represented as shown in Figure 8, fitting into the MWIR spectral range of interest.

The model can switch between a linear variable filter and a fixed-line pass band filter to be the spectral differentiator in the sensor assembly. The manufactured LVF’s have a spectral range of 2.9 to 4.8 µm, with a pass band width of 1.5% and peak transmission of 70%. The LVF transmision is modelled as a Gaussian distribution around the central wavelength of interest. The transmittance at the centre of each of the detector element is averaged to yield a spectrum shown in Figure 9—with lower average transmittance, as expected, at the lower end of the wavelength range and higher average transmittance at upper wavelengths [35].

The fixed pass band filters are modelled to match ideal band pass characteristics suited for improved gas sensitivities, as well as to emulate available commercial fixed filter specifications [36].

The reference channel filter’s central wavelength is chosen from modelling design, to be as close to the signal channel filter’s—as shown in Figure 10. This ensures temperature shifts have minimal effects on the performance of the filters and the detector output.

When using a fixed pass band optical filter, issues such as light leakage onto adjacent channels, reduced transmittances at lower wavelengths, inherent to an LVF, are significantly reduced using fixed-line pass band filters over the detector elements, providing enhanced gas sensitivity.

The Source–Detector–Filter product is then modelled to consolidate accurate placement of the filter and provide baseline model performance before gas introduction as shown in Figure 11.

## 3. Results

### 3.1. Absorption Modelling Calculation

The gas cell is then modelled by calculating transmittance through the gas of interest. A total of 600 data points is chosen within the detector wavelength range and the product of transmittance and source–detector–filter is calculated at each of these points and integrated over the wavelength range. The transmittance is calculated using Beer–Lambert’s law, which determines the relation between absorbance and concentration of the infra-red absorbing medium, which, in this case, are the gas molecules of interest. Methane detection is explored in detail, in the context of this paper, and water vapour is chosen as the back-filling gas. Transmittance, which is the ratio of the intensity of IR after absorption to the intensity of IR after absorption, is estimated as:(5)$$T_{\mathrm{CON}}=e^{-\left(\alpha_{gas_{1}}\right)\times c1\times L_{\mathrm{cell}}}\times e^{-\left(\alpha_{gas_{2}}\right)\times c2\times L_{\mathrm{cell}}}$$
where α_gas1_ and α_gas2_ are the molar absorptivities of the target gas and the back gas, respectively. L_cell_ is the optical path length of the gas cell and c1, c2 are the gas concentrations of the target and the back gas.

The effective detected signal generated can thus be estimated over the entire wavelength range of detection as:(6)$$S=O\times \sum \left(T\times F\times S\times D\right)$$
where O is the optical throughput of the sensor

T is TCON, the transmittance through gas

F is the effective optical filter transmittance

S is the net source spectral strength

D is the net detectivity of selected detector channel

The signal strength is calculated in a similar manner, for the chosen detector reference channel as well. The Signal to Reference ratio is normalized and plotted against the gas concentration as shown in Figure 12 and the slope of the curve is estimated to determine the sensitivity of gas detection. The effects of temperature on the sensitivity measurements can also be seen, which makes the specifications of the chosen signal and reference filters even more critical for maintaining an acceptable level of deviation.

The gas sensitivity to Methane was modelled as follows in Table 1:

The integration time or the time for making a precise single measurement is dependent on the number of measurement pairs chosen from the peak and trough of the detector response pulse, the total number of pulses (the frequency of operation of the micro-hotplate). The slope of the Signal to Reference curve is then taken and compensated for quantization noise, to determine the measurement accuracy of the sensor to a specific gas, in this case, methane.

Similarly, other gases of interest such as carbon dioxide, carbon monoxide, nitrogen dioxide, sulphur dioxide, ammonia and benzene are modelled and the signal and reference channel filter characteristics are varied to get down to detection accuracies of the required levels for each of the gas.

### 3.2. Power Consumption

The multi-gas sensor can be operated at a very low power in the order of milli Joules, depending on the frequency of operation of the source and the number of data samples from each pulse required to maintain an acceptable level of accuracy. For a 1 Hz operation of the sensor, the energy consumed for toggling the hot plate ON and OFF can be as low as 13 mJ/s. This is excluding the current drawn by the analog circuitry during the microcontroller’s IDLE state and ADC (Analog-to-Digital Converter) activation stages. For measurement of individual gases, the total energy per measurement depends on the number of samples from the detector pulse used for making a meaningful measurement, which would in turn, depend on the total integration time for measurement. Table 2 shows the average power consumption charts for gas measurements and Figure 13 illustrates the control of pulse-width as an efficient method to reduce energy consumed per measurement.

### 3.3. Gas Concentration Tests

Gas concentration tests were conducted within a custom-made gas chamber. Using a mass flow controller and nitrogen as the inert gas, experiments were conducted under a constant temperature. The gas chamber volume was chosen to be small, per design, to establish lower stabilization times after the introduction of gas.

Mass flow meters enabled gas mixing and the ratio with the inert gas, in this case, Nitrogen, was used to control the percentage/ppm levels of the target gas to be tested. Carbon dioxide, methane and carbon monoxide were tested using the setup shown in Figure 14.

Using assembled fixed-line filter array as the spectral differentiator (as shown in Figure 15), the following results were obtained:
(a)Methane was detectable at 100 ppm, 500 ppm and 25,000 ppm (below lower explosive limit)—but longer sampling times and faster sampling techniques are required as these improve detection by compensating for the noise fluctuations in the system. Figure 15 shows how the PbSe detector output indicated 2.5% Methane detection.In addition to methane, the following performance for CO_2_ and CO observed:(b)CO_2_: Very good detection at 50 ppm, 100 ppm and 170 ppm.(c)CO: Tested at 50 ppm and tests did not show evident detection due to lowered S/N levels in the system. CO detection is more difficult due to the centre wavelength at 4.65 µm lying on the falling edge of the lead selenide detector *D**—lambda curve.

### 3.4. Temperature Sensitivity

It was observed during gas sensitivities modelling that the bandwidth and central wavelength of the fixed filters used are critical to maintaining an acceptable level of temperature offset between the signal and reference channel output signals. The reference channel is designated mainly to account for temperature-related corrections. Inherent output shifts due to the nature, quality and position of the filter used are disruptive and should be corrected by strategically designing the signal and reference filters for each gas. The importance of this is illustrated in carbon monoxide detection modelling, as shown in Figure 16a,b below.

The gas detection accuracies for both Design A and Design B (Figure 16a,b) were modelled at room temperature 25 °C and an increased temperature to 60 °C. As observed from the following Figure 17 and Figure 18, the temperature effects on the fixed pass band filters and its resultant impact on the gas sensitivities are adverse when using wider commercial pass band filters, which did not have optimal reference central wavelengths.

The results showed that for each gas, depending on the nature, shape and spread of the infrared absorption spectrum, the fixed pass band filters should be designed with precision, to ensure the minimum temperature-related response between the signal and the reference detector channel.

For the six gases of interest, the specifications drawn from the model results are tabulated in Table 3 below.

### 3.5. Varying Pathlengths and Optical Throughputs

For detection of moderately high to very high concentrations of gases which possess strong IR-absorption characteristics such as CO_2_ and CH_4_, the path length of the cell needs to be designed carefully to avoid saturation of the gas cell and avoid 100% absorption. Impacts of varying path lengths and effects of using designs with different optical throughputs on gas detection, as well as the Signal-to-Noise ratio (S/N) of the sensor, were investigated by modelling for CO_2_ as shown in Figure 19 below, for varying path lengths and optical throughputs:

For CH_4_ gas, the change of modelled output for various concentrations are compared for two different path lengths 60 mm and 90 mm, as shown in Figure 20. Noise levels remaining the same, it is seen that the S/N ratio increased almost 1.4 times for the 90 mm path length model, when compared to the 60 mm version, at low concentrations typical for methane detection.

### 3.6. Measurement Accuracy for Various Gases

The measurement accuracy for the various gases of interest are summarized in Table 4.

## 4. Conclusions and Future Work

In this work, we have demonstrated a novel NDIR-based gas sensor design with a folded optics arrangement providing a 60 mm pathlength with a single broadband light source/detector array optopair providing efficient simultaneous detection of up to six gases (CO_2_, CH_4_, CO, NO_2_, H_2_S, NH_3_) with detection levels aligned with commercial use. The infrared source/detector array optopair, optics, and processing electronics were characterized and modelled to understand limitations and areas for improvements in signal acquisition, temperature sensitivities, signal corrections including processing techniques, integration time, calibration algorithms, optical filter spectral characteristics. Experiments have demonstrated that the micro-hotplate source can be driven using very short current pulses, effectively reducing power consumption into the mJ range. Modelling indicates energy consumption per measurement is typically 13 mJ for a ±3% gas detection measurement accuracy.

A C++ based data acquisition system and a Python-based post-processing tool was developed and used to extract of the microcontroller data from the sensor using a serial USART interface and real time data acquisition. The signal processing is presently done outside the sensor unit, which restricts the pulsing frequency of the hot plate source due to the slowness of acquisition. Future work will focus on enhancing measurement accuracy by improving/shaping the source pulsing characteristics. Also, a next stage to this work will be to incorporate the signal processing into the microcontroller, making the sensor completely stand-alone and incorporating temperature calibration and compensation algorithms into the firmware.

## Figures and Tables

**Figure 1 sensors-20-03843-f001:**
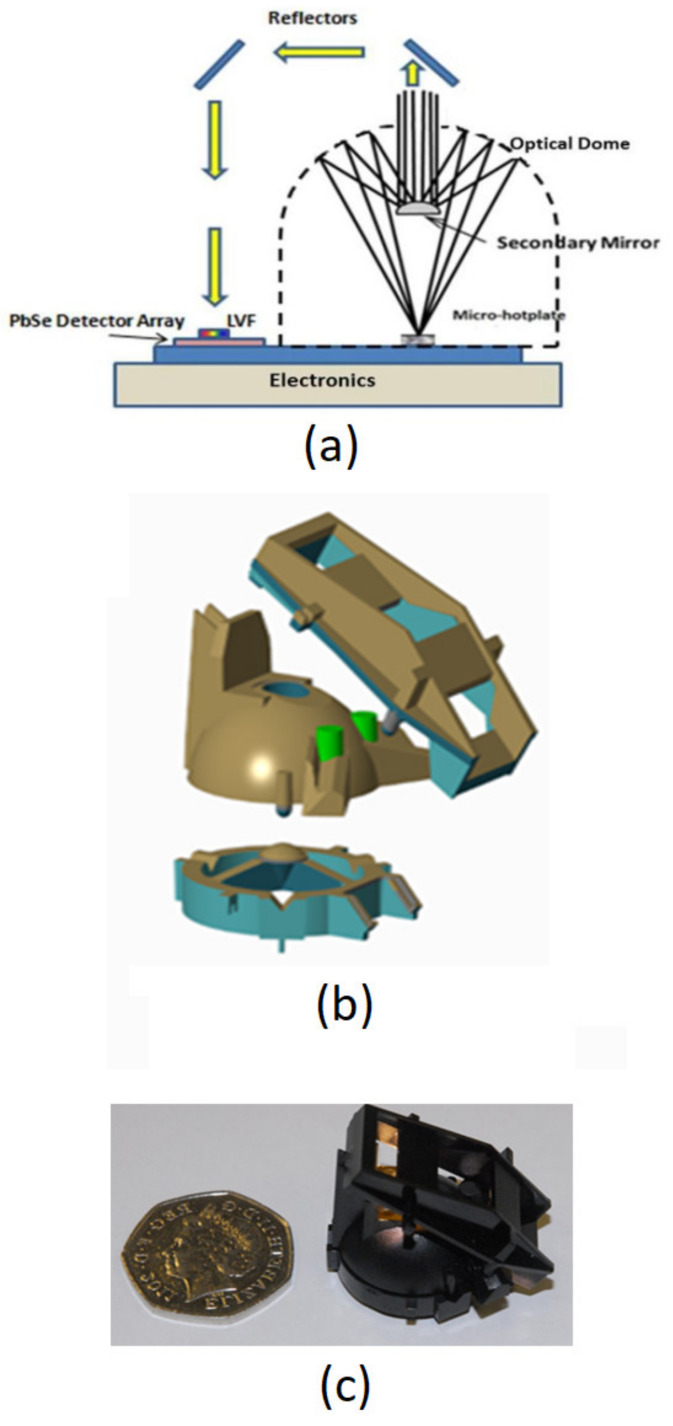
(**a**) Schematic of the multi-gas sensor optics; (**b**) exploded view of the three-piece optical assembly; (**c**) assembled injection moulded optics (UK 50 pence piece shown for scale—23 mm diameter).

**Figure 2 sensors-20-03843-f002:**
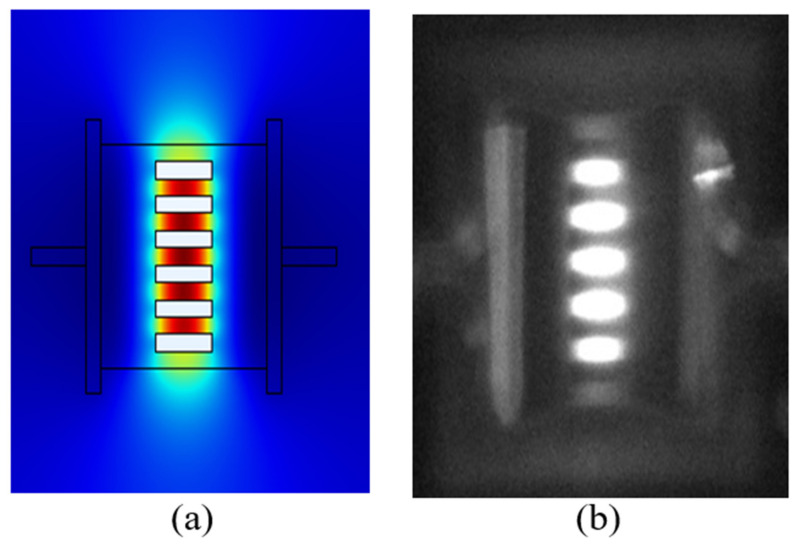
(**a**) Modelled Micro Hotplate showing solid contours for temperature of the microheater with PolySi layer built on a Si3N4 substrate (**b**) IR-camera image of the actual Micro Hotplate.

**Figure 3 sensors-20-03843-f003:**
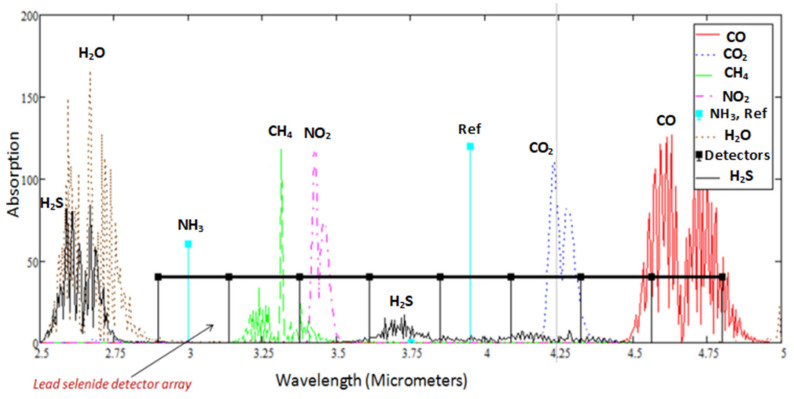
Lead selenide detector alignment with gas absorption spectra.

**Figure 4 sensors-20-03843-f004:**
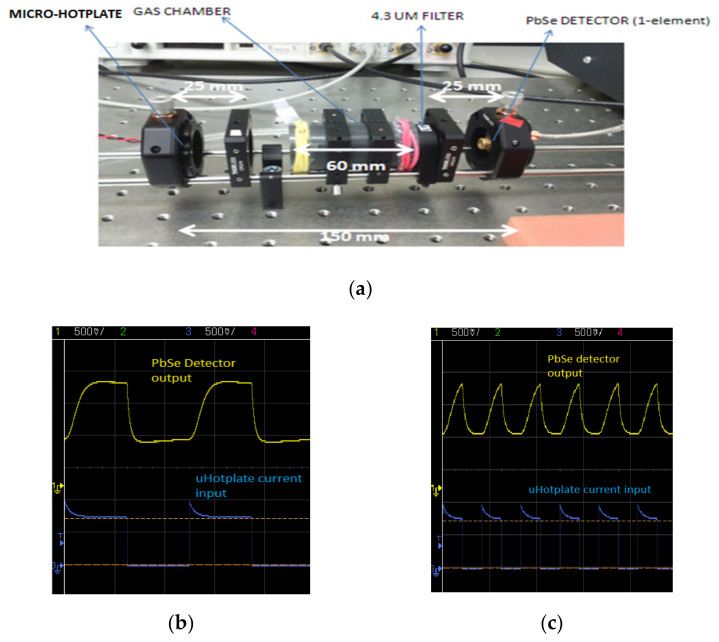
(**a**) Demonstrator setup for gas detection with 25 mm optical lenses, (**b**) output with the hotplate driven with a 4 Hz square wave, and (**c**) output with hotplate driven with 12 Hz square wave.

**Figure 5 sensors-20-03843-f005:**
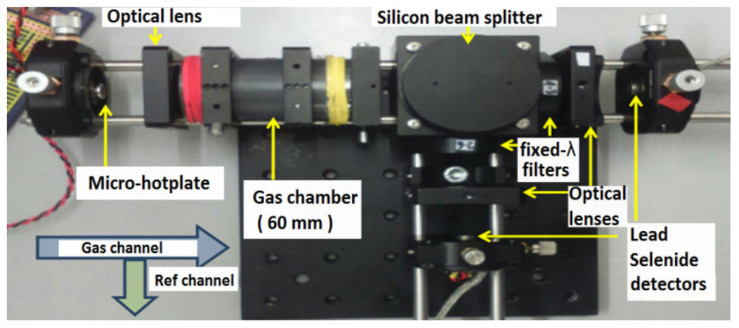
The demonstrator utilizing two single-element lead selenide detectors for the gas and reference channels.

**Figure 6 sensors-20-03843-f006:**
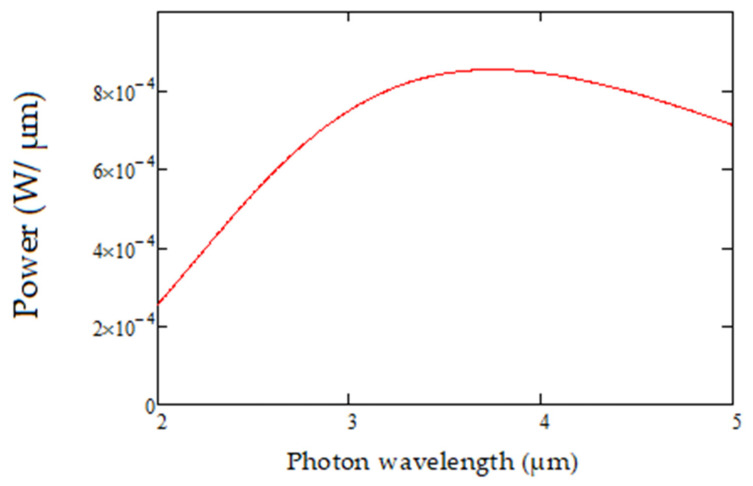
Source spectral radiant power for the micro-hotplate source.

**Figure 7 sensors-20-03843-f007:**
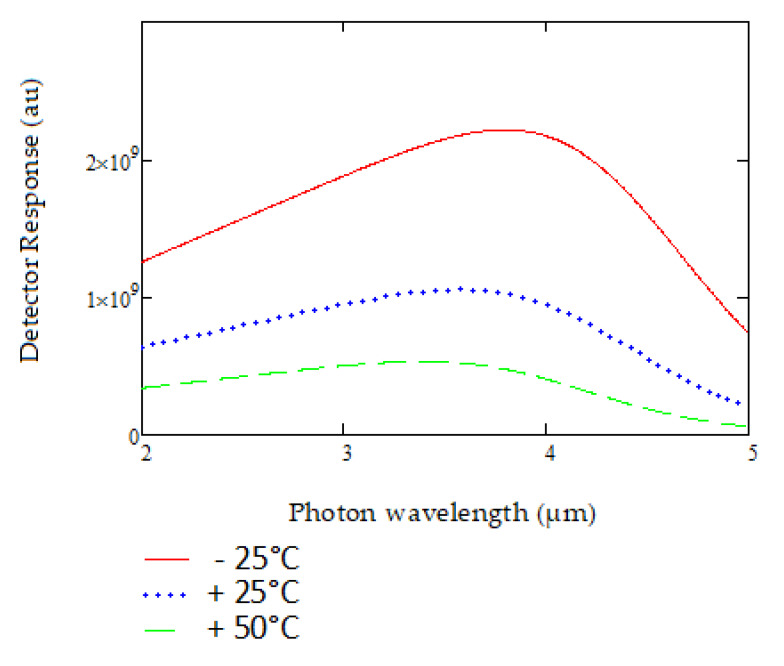
Lead selenide detector response modelled at different temperatures in the 2 to 5 µm range.

**Figure 8 sensors-20-03843-f008:**
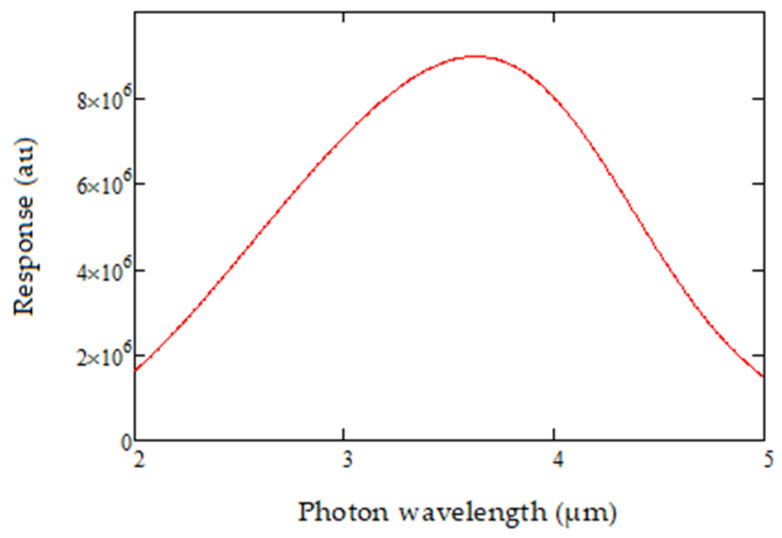
Source-Detector product over the detection range of interest.

**Figure 9 sensors-20-03843-f009:**
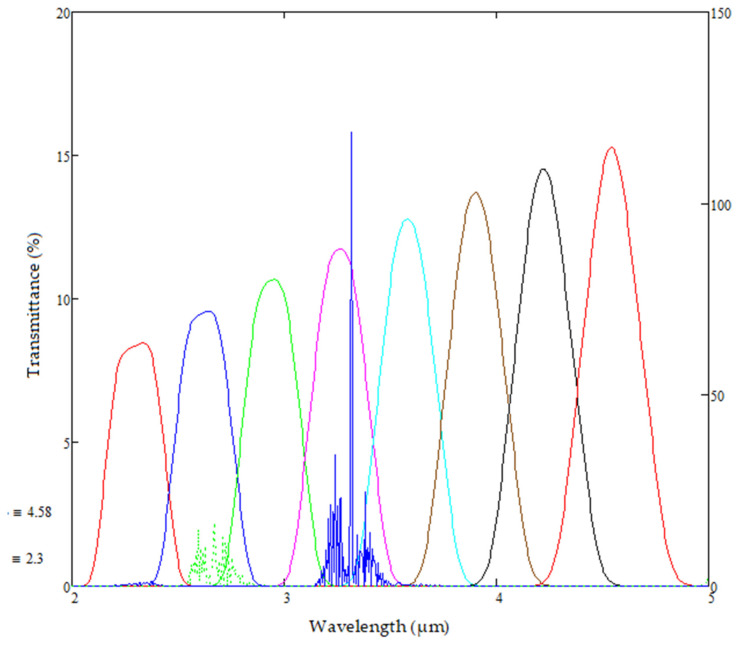
Modelled average spectra over each element of the lead selenide detector array after placement of linear variable filter.

**Figure 10 sensors-20-03843-f010:**
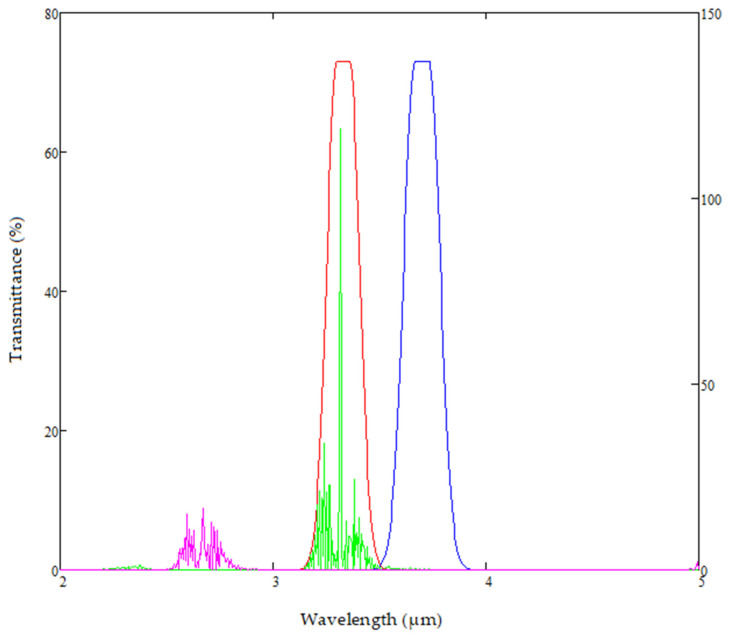
Signal and Reference filter placement using fixed pass band optical filters.

**Figure 11 sensors-20-03843-f011:**
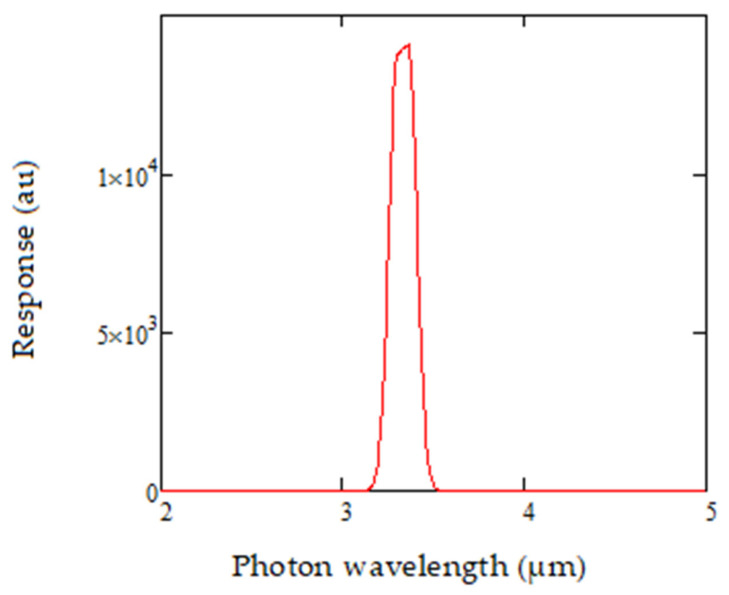
Source–Detector–Filter product for methane gas detection.

**Figure 12 sensors-20-03843-f012:**
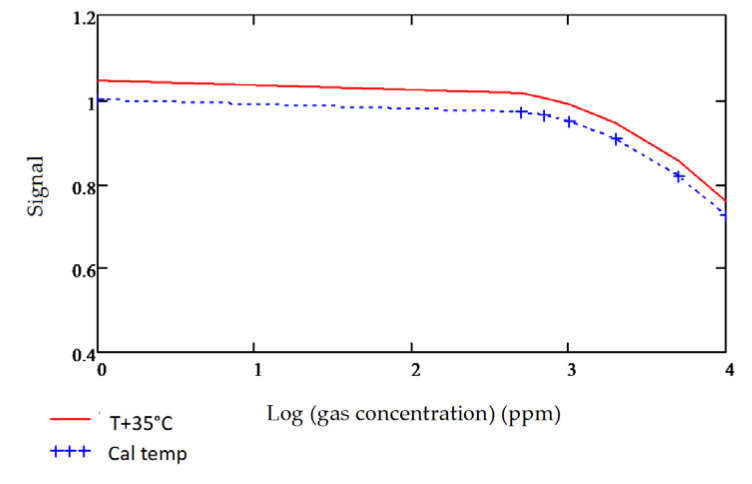
Source–Detector–Filter product for methane gas detection.

**Figure 13 sensors-20-03843-f013:**
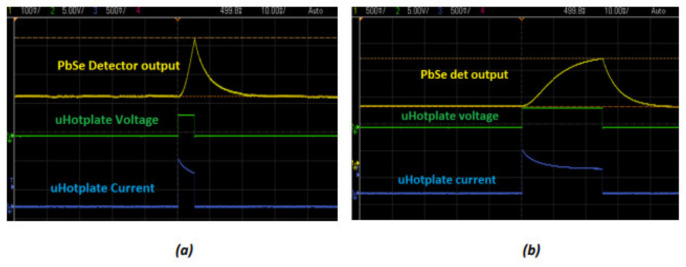
(**a**) System output with hotplate pulsed at 5 ms and (**b**) System output with hotplate pulsed at 25 ms.

**Figure 14 sensors-20-03843-f014:**
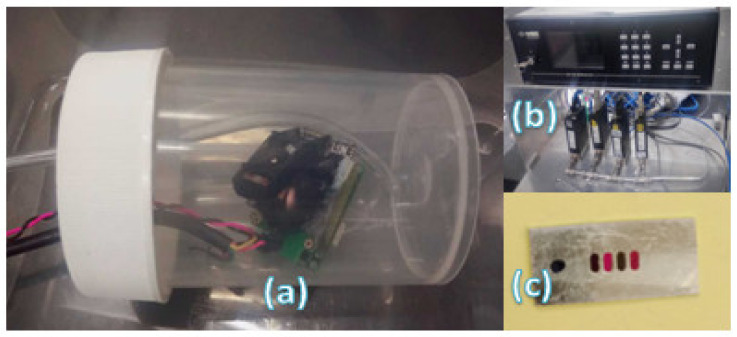
(**a**) Gas sensor custom-built chamber; (**b**) Mass Flow Controllers for gas mixing; (**c**) Aluminium-based filter array holders for testing fixed-line passband filters.

**Figure 15 sensors-20-03843-f015:**
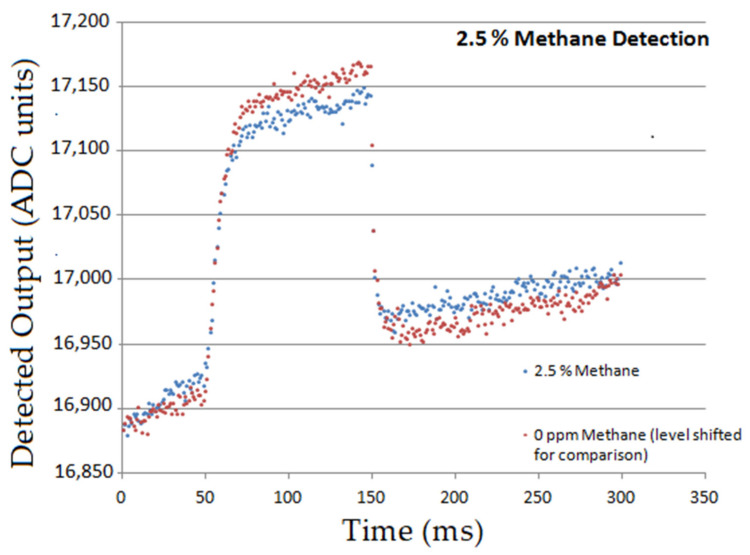
PbSe detector output demonstrating 2.5% Methane detection.

**Figure 16 sensors-20-03843-f016:**
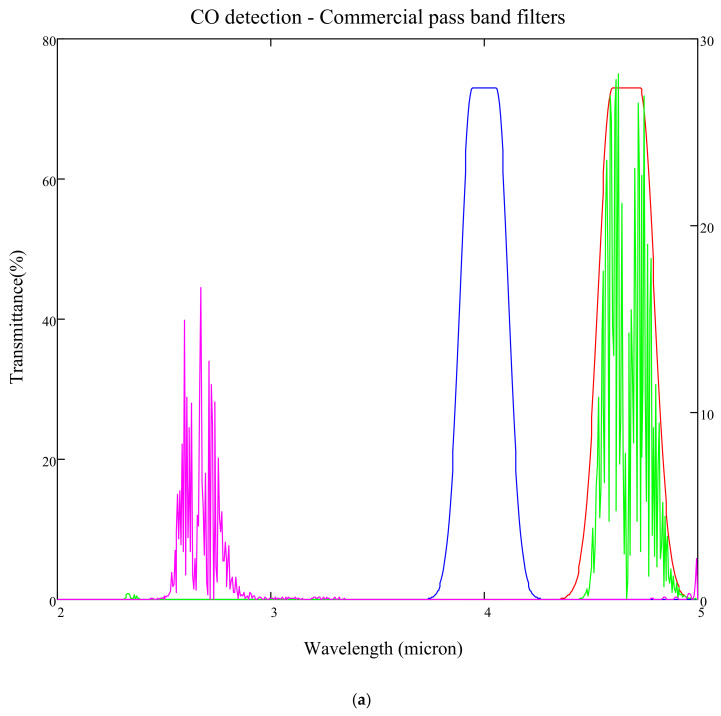
(**a**) Design A: CO filter transmission using 480 nm wide commercially available signal [37] and reference filters. (**b**) CO filter transmission using 100 nm wide in-house designed signal filter and reference filters [33].

**Figure 17 sensors-20-03843-f017:**
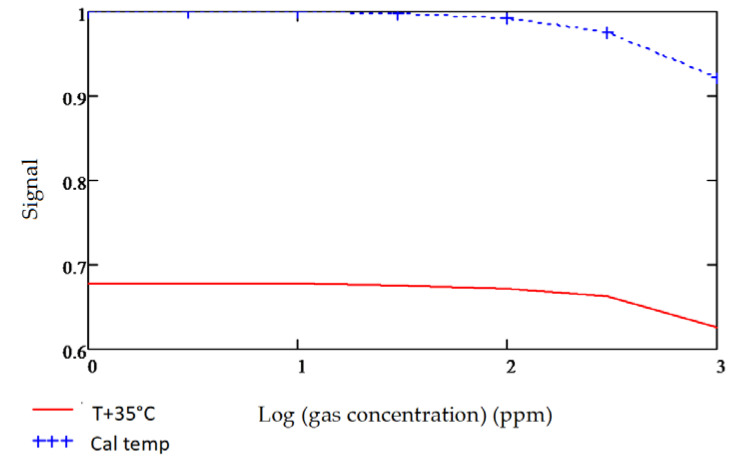
Design A: Detected lead selenide output with commercial signal and reference filters.

**Figure 18 sensors-20-03843-f018:**
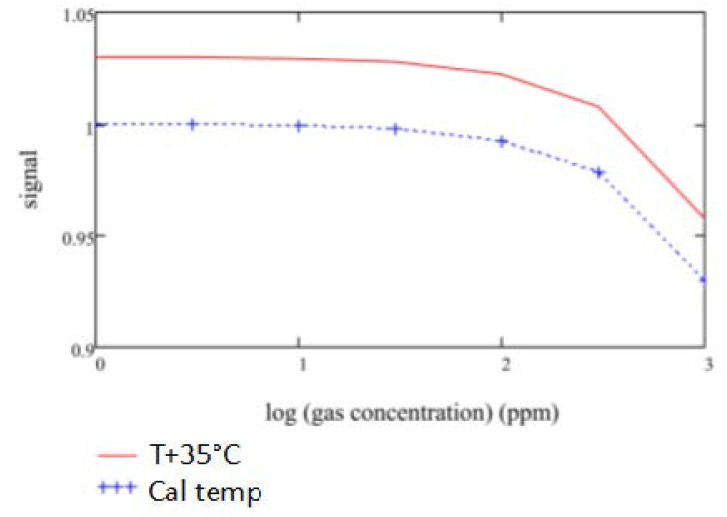
Design B: Detected lead selenide output with ideally designed signal and reference filters.

**Figure 19 sensors-20-03843-f019:**
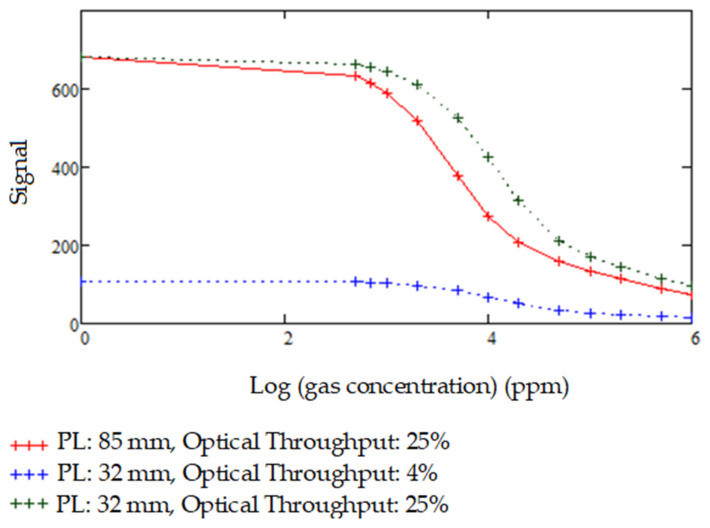
Detectable signal modelled for different path length–optical throughput combinations for CO2 gas concentrations upto 1,000,000 ppm.

**Figure 20 sensors-20-03843-f020:**
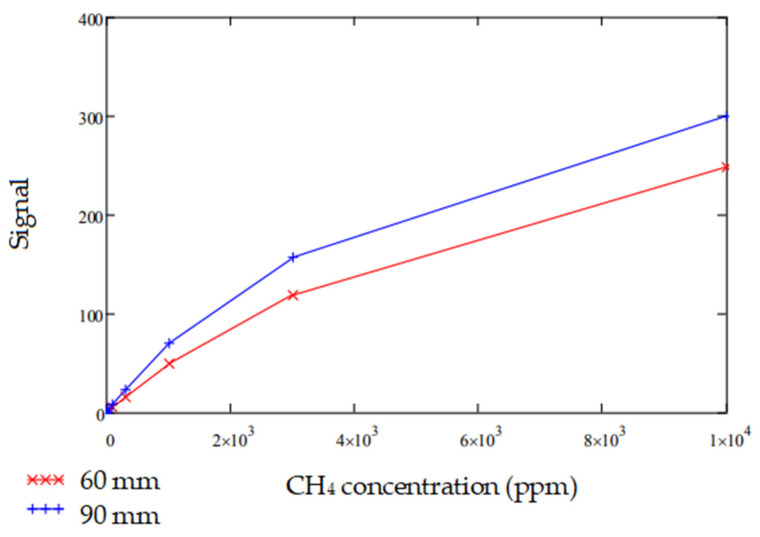
Change in signal modelled for different path lengths 60 mm and 90 mm for CH_4_ gas concentrations below 10,000 ppm.

**Table 1 sensors-20-03843-t001:** Modelled methane gas sensitivity values for different concentration ranges.

Measurement Range	Accuracy of Measurement
0 to 100 ppm	±25 ppm of measured reading
100 to 1000 ppm	±1% of measured reading
21,000 to 1,000,000 ppm	±1.1% of measured reading

**Table 2 sensors-20-03843-t002:** Power requirements for a single measurement of different gases.

Gas	Samples Required	Pulses Required	Total Time (s)	Total Energy (mJ)	Average Power Consumption (mW)
CO_2_	100	1	1	13	13
CH_4_	100	1	1	13	13
CO	100,000	1000	1000	13,000	13
NO_2_	100,000	1000	1000	13,000	13
H_2_S	1,000,000	10,000	10,000	13,000	13
NH_3_	1,000,000	10,000	10,000	13,000	13

**Table 3 sensors-20-03843-t003:** Modelled ideal fixed filter specifications for both signal and reference filters for the various gases of interest.

(**a**): CO				
**Filter**	**Central-λ (um)**	**FWHM of Peak (um)**	**FWHM of Slope (um)**	**Transmission %**
Signal	4.6	1	0.5	73
Reference	4.665	0.5	0.5	73
(**b**): NO_2_				
**Filter**	**Central-λ (um)**	**FWHM of Peak (um)**	**FWHM of Slope (um)**	**Transmission %**
Signal	3.43	1	0.5	73
Reference	3.495	0.5	0.5	73
(**c**): NH3				
**Filter**	**Central-λ (um)**	**FWHM of Peak (µm)**	**FWHM of Slope (µm)**	**Transmission %**
Signal	3.0	1	0.5	73
Reference	3.065	0.5	0.5	73
(**d**): H_2_S				
**Filter**	**Central-λ (um)**	**FWHM of Peak (µm)**	**FWHM of Slope (µm)**	**Transmission %**
Signal	3.98	1	0.5	73
Reference	3.92	0.5	0.5	73
(**e**): CH_4_				
**Filter**	**Central-λ (um)**	**FWHM of Peak (µm)**	**FWHM of Slope (µm)**	**Transmission %**
Signal	3.33	3.0	2	73
Reference	3.5	1.6	2	73
(**f**): CO_2_				
**Filter**	**Central-λ (um)**	**FWHM of Peak (µm)**	**FWHM of Slope (µm)**	**Transmission %**
Signal	4.26	3.0	2	73
Reference	4.6	1.6	2	73

**Table 4 sensors-20-03843-t004:** Accuracy of gas measurements in ±percentages of the measured value or ±ppm of measured ppm—for gases of interest and total measurement integration times.

Target Gas ppm Range	CO_2_	CH_4_	CO	NO_2_	H_2_S	NH_3_
0 to 1000	25 ppm	25 ppm	8%	9.9%	11.9%	12.6%
1000 to 10,000	1%	1%	1.39%	3.12%	5.6%	7.6%
10,000 to 1,000,000	3%	1.1%	0.8%	1%	2.3%	4.8%
Integration time (s)	1 s	1 s	4 min	4 min	42 min	43 min

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
