# Peer review of "Miniaturised Infrared Spectrophotometer for Low Power Consumption Multi-Gas Sensing"

_sensors, 2020, doi:10.3390/s20143843_

Round 1

Reviewer 1 Report

This paper presents the design and implementation of a miniaturized spectrophotometer as a mid-infrared based multi-gas detector, providing a novel, standalone, low power, accurate gas sensor especially suitable for the autonomous sensor applications (e.g. as an IoT sensor node). Simulations and characterization experiments demonstrated it can detect gas effectively and efficiently. In general, it is a good paper well-fit the scopes of this journal.
There are some minor issues to be improved:
1. the conclusion should be to rewrite to reflect what achieved rather than what have done. Also, future work should be moved to the discussion.
2. the qualities of the figures should be improved.
Fig 8 should be removed since it repeated in Fig. 9;
Considering a more professional way to present the data: can Fig.18 combined to Fig. 19 as one?
Fig. 15 should be better used in Section 3.2
Fig. 2 is not mentioned in the context
3. remove unnecessary data: communication data rate is reported in P5 line 150. If this rate is not directly liked to sensor performance such as minimum response time, then it should not be included. Otherwise more explanation is required.
4. Some format issues:
P9 line 228 needs space before [ ];
Both Fig XX and figure XX are used in the main body

Author Response

Reviewer 1:

This paper presents the design and implementation of a miniaturized spectrophotometer as a mid-infrared based multi-gas detector, providing a novel, standalone, low power, accurate gas sensor especially suitable for the autonomous sensor applications (e.g. as an IoT sensor node). Simulations and characterization experiments demonstrated it can detect gas effectively and efficiently. In general, it is a good paper well-fit the scopes of this journal. 

There are some minor issues to be improved:

Please refer to an updated attached paper with reference to line numbers (as per reviewer comments) responses below.

  1. the conclusion should be to rewrite to reflect what achieved rather than what have done. Also, future work should be moved to the discussion.

Response:

Conclusion rewritten to reflect what has been achieved, also future work incorporated into a renamed section conclusions and future work.

Discussion section renamed conclusions and future work (see line 422, section 4).

  1. the qualities of the figures should be improved. 
    Fig 8 should be removed since it repeated in Fig. 9;
    Considering a more professional way to present the data: can Fig.18 combined to Fig. 19 as one?
    Fig. 15 should be better used in Section 3.2
    Fig. 2 is not mentioned in the context

Response:

Figures have been improved with increased font size axis titles and headings removed to provide a more professional appearance. Now consistent font and layout for all figures.

Figure 8 now removed as repeated in figure 9.

Figure 15 has been replaced in section 3.2.

Figure 18 and 19 are required as they show both bandpass filters used for the project.

Figure 18 commercially purchased filter - reference 37

Figure 19 in house designed filter – reference 33

All figures now mentioned in the body of text.

  1. remove unnecessary data: communication data rate is reported in P5 line 150. If this rate is not directly liked to sensor performance such as minimum response time, then it should not be included. Otherwise more explanation is required. 

Response:

Agreed, not directly related and removed as suggested.

  1. Some format issues:
    P9 line 228 (new line number as per attached is 232)  needs space before [ ];
    Both Fig XX and figure XX are used in the main body

Response:

Reference spacing and figure issues have been addressed and changed. ‘Figure’ or ‘figure’ has been implemented throughout.

Reviewer 2 Report

The paper presents an overview of the authors`work on a miniaturized spectrophotometer for multi-gas sensing. Since the topic is of high current interest, the paper is relevant and of high interest to the audience.

There are some comments:

-In the introduction the authors give some overview about NDIR sensors. But they miss to mention that this technique is also applied for the development of completely integrated gas sensors (There is a lot of work towards integrated Mid-Infrared gas sensors going on currently, including an excellent review by Popa et al. from last year.) In the context of "miniaturized gas sensors" this should be mentionned.

-In the text, the authors describe 3 different systems. "prototype", "initial sensor" and "assembled sensor". (Fig. 4, 5, 6) It is not always clear, how the experiments were performed. This should be clearly stated and discussed, for a better understanding of the reults.

-The micro-hotplate is described rather superficially. It is built from poly-Si. Is it doped? How is it contacted?
The authors mention a further publication with more details. They should put a reference, to where this publication is planned.

-Captions of figure2: Is the difference between (b) and (c) the fact that in (b) it is sandwitched between the substrates and (c) it is the hotplate alone?(Still it must be on a substrate, isn´t it?). Or is there a difference between "heat camera" and "IR-camera" (I thought that´s the same)

-line88: referecne to "section C" propably refers to Sec. 2.3!??

-line170: In the formula, the first numerator reads: "2 x pi x h x r^2". Shouldn´t it be "2 x pi x h x c^2"?

-line 185 and below: In the text it refers to D* and in the formulas to "Dstar". It is clear, but would be more readable, if the same letters are used.

-Equ.(3): The speed of light is labeled "c_SI", while above (in eq. 1) it was labeled "c". Is there some reason or subtle difference?

-line 208,217,226...: Reference to the figures are wrong. ("Figure 7" should be "Figure 8" etc...)

-line 370: The text mentions "For the THREE gases of interest...", however, in the table six gases are discussed.

-line 402: Measurements with 60 and 90mm were measured. On which system, these measurements were performed, and how was the path length variation realized in detail?

-chapt. 2.5: More details of the design of the system would be highly interesting, since this is the key system, justifying the claim of a "miniaturized" spectrometer in the title of the paper. How are the optical dimensions, materials, coatings?

Author Response

Reviewer 2:

The paper presents an overview of the authors`work on a miniaturized spectrophotometer for multi-gas sensing. Since the topic is of high current interest, the paper is relevant and of high interest to the audience.

There are some comments:

-In the introduction the authors give some overview about NDIR sensors. But they miss to mention that this technique is also applied for the development of completely integrated gas sensors (There is a lot of work towards integrated Mid-Infrared gas sensors going on currently, including an excellent review by Popa et al. from last year.) In the context of "miniaturized gas sensors" this should be mentionned.

Response:

Please refer to an updated attached paper with reference to line numbers in responses below.

Now amended to include integrated and miniaturised gas sensor aspect inclusion of Popa et al reference.

-In the text, the authors describe 3 different systems. "prototype", "initial sensor" and "assembled sensor". (Fig. 4, 5, 6) It is not always clear, how the experiments were performed. This should be clearly stated and discussed, for a better understanding of the reults.

Response:

 References to the miniaturised photospectrometer in the main body of the text have been replaced with the single description of “sensor”. The transmissive optics breadboard setup is described as a “demonstrator”. The lead selenide array and single element components are describes as “detectors” and the infrared hot-plate is the “source”.

-The micro-hotplate is described rather superficially. It is built from poly-Si. Is it doped? How is it contacted?
The authors mention a further publication with more details. They should put a reference, to where this publication is planned.

Response:

‘Non-uniform voltage equipotential lines near edges of the slits cause a greater pull of electrons and consequently, higher current density. This results in increased temperature between the slits. The micro-hotplate length is customized to 0.6mm length, providing magnification to 6mm at the detector array (see section 2.3). The broadband spectrum produced by the micro hotplate provides the required signal spread over all elements of the lead selenide infrared detector array. The hotplate is operated in a constant voltage mode, using a circuit developed specifically to provide a battery compatible electromotive force of < 5V.’

The position of the aluminium electrodes in Figure 2(a) is highlighted in the text, lines 82-87.

-Captions of figure2: Is the difference between (b) and (c) the fact that in (b) it is sandwitched between the substrates and (c) it is the hotplate alone?(Still it must be on a substrate, isn´t it?). Or is there a difference between "heat camera" and "IR-camera" (I thought that´s the same)

Response:

Figure 2(a) and 2(b) are now referenced in the text. The image previously described as derived from a heat camera is not required and did not support a meaningful discussion - deleted.

Future publication planned, journal being considered is Sensors and Actuators A: Physical.

-line88: referecne to "section C" propably refers to Sec. 2.3!??

Response:

Noted and changed accordingly, see line 100.

-line170: In the formula, the first numerator reads: "2 x pi x h x r^2". Shouldn´t it be "2 x pi x h x c^2"?

Response:

Yes, your comment is correct, thank you for pointing this out. Changed to the correct formula, see Equation 1 line 176.

-line 185 and below: In the text it refers to D* and in the formulas to "Dstar". It is clear, but would be more readable, if the same letters are used.

Response:

Agreed, all ‘Dstar’ changed to ‘D*’, see formula 2 and 4, lines 193 and 208.

-Equ.(3): The speed of light is labeled "c_SI", while above (in eq. 1) it was labeled "c". Is there some reason or subtle difference?

Response:

No difference between the two. Is now corrected and ‘c’ is used, line 198.

The Mathcad model used adds ‘si’, now removed for consistency throughout.

-line 208,217,226...: Reference to the figures are wrong. ("Figure 7" should be "Figure 8" etc...)

Response:

Figure captions and in text figure references corrected throughout.

-line 370: The text mentions "For the THREE gases of interest...", however, in the table six gases are discussed.

Response:

This is now six gases of interest in the text, now ties in with the six gases in table 3, as can be seen in lines 375.

-line 402: Measurements with 60 and 90mm were measured. On which system, these measurements were performed, and how was the path length variation realized in detail?

Response:

The modelling work (PC based) used two path lengths (60mm and 90mm) for comparative analysis.  All experimental measurements on the sensor used a 60mm path length.

-chapt. 2.5: More details of the design of the system would be highly interesting, since this is the key system, justifying the claim of a "miniaturized" spectrometer in the title of the paper. How are the optical dimensions, materials, coatings?

Response:

Extra detail added to section 2.1 and 2.2, also justification of miniaturised spectrophotometer in section 2.1.

Optical dimensions, materials and coatings now included.

Reviewer 3 Report

This paper “Miniaturised infrared spectrophotometer for low power consumption multi-gas sensing” focused on a concept, design and practical implementation of a miniaturized spectrophotometer based on non-dispersive infrared (NDIR) spectroscopy.

The paper is well constructed, and the experimental investigation is performed deeply.

But the Introduction section needs revise. The author compared NDIR with chemical based gas sensors. It is unfair. NDIR is an optical method. So it should make a comparison with some other optical techniques, such as TDLAS, PAS, QEPAS, and QEPTS.

Some suggestions are following: “Multi-resonator photoacoustic spectroscopy”. Sensors and Actuators B. 2017, 251, 632-636; “In plane quartz-enhanced photoacoustic spectroscopy.” Applied Physics Letters. 2020, 116, 061101; For QEPTS, you can google to find the reference.

Author Response

Reviewer 3:

This paper “Miniaturised infrared spectrophotometer for low power consumption multi-gas sensing” focused on a concept, design and practical implementation of a miniaturized spectrophotometer based on non-dispersive infrared (NDIR) spectroscopy.

The paper is well constructed, and the experimental investigation is performed deeply.

But the Introduction section needs revise. The author compared NDIR with chemical based gas sensors. It is unfair. NDIR is an optical method. So it should make a comparison with some other optical techniques, such as TDLAS, PAS, QEPAS, and QEPTS.

Some suggestions are following: “Multi-resonator photoacoustic spectroscopy”. Sensors and Actuators B. 2017, 251, 632-636; “In plane quartz-enhanced photoacoustic spectroscopy.” Applied Physics Letters. 2020, 116, 061101; For QEPTS, you can google to find the reference.

Response:

Please refer to an updated attached paper with reference to line numbers in responses below.

Agree with all reviewer comments. Introduction now modified to include other techniques mentioned TDLAS, PAS, QEPAS, and QEPTS. Additional references added which reviewer highlighted. NDIR compared with chemical based gas sensors relevant as NDIR solves performance shortfalls in chemical based gas sensors as now described within the introduction.

Reviewer 4 Report

Muhiyudin et.al present the design of a NDIR sensor. While the description of the experimental work is acceptable there is no novelty whatsoever in the manuscript. All of the componented as well as the combination have been presented in the past.

The manuscript appeaers to be a design study and not a scientific paper. Apart from this fundamental flaw, the state of the art in NDIR sensing devices is not discussed at a sufficient leven.

I am afrain that my vote has to be to reject this manuscript for sensors.

Author Response

Muhiyudin et.al present the design of a NDIR sensor. While the description of the experimental work is acceptable there is no novelty whatsoever in the manuscript. All of the componented as well as the combination have been presented in the past.

The manuscript appeaers to be a design study and not a scientific paper. Apart from this fundamental flaw, the state of the art in NDIR sensing devices is not discussed at a sufficient leven.

I am afrain that my vote has to be to reject this manuscript for sensors.

Response:

With respect we completely disagree with reviewer fours comments for the following reasons.

The described work includes both modelling and measured performance as clearly articulated in the paper and supported by scientific reasoning and as such is a scientific paper and not a design study as alleged by reviewer 4.

The associated patented optical configuration - reference 24 (Inventors Gibson and Waddell [authors in this submitted publication] Patent – Miniaturised Infrared Spectrophotometer) – indicates  and supports the novelty of the described configuration.  Moreover the described gas sensor has simultaneous multigas sensing capability (up to six gases) compared to other NDIR configurations with no more than two in the one gas sensing unit.

Moreover use of linear variable filters combined with linear infrared detector  to provide infrared spectrometry associated patent reference 32 (Song, Gibson, Hutson [authors in the submitted publication]- Patent Apparatus & Methods for depositing Variable Interfenrece Filters) also indicates further novelty in the described multi-gas gas sensor.

State of the art in NDIR is comprehensively discussed in the introduction with supporting references.

The lack of specific evidence relating to the manuscript to support reviewer 4 brief comments is unacceptable. More detail is required to substantiate the reviewers comments. It is obvious from reviewer 4`s comments there is a lack of understanding of the submitted manuscript content and relevance.

Round 2

Reviewer 2 Report

The authors adequately responded to my questions and the paper is ok for publication, in my opinion.

There are some formatting and minor spelling errors.